# Krüppel-Like Factor 1: A Pivotal Gene Regulator in Erythropoiesis

**DOI:** 10.3390/cells11193069

**Published:** 2022-09-29

**Authors:** Cristian Antonio Caria, Valeria Faà, Maria Serafina Ristaldi

**Affiliations:** Istituto di Ricerca Genetica e Biomedica del Consiglio Nazionale delle Ricerche (IRGB-CNR), Cittadella Universitaria, SS 554 Bivio per Sestu km 4,5, 09042 Monserrato, Italy

**Keywords:** KLF1, erythropoiesis, haemoglobin switching, human mutations

## Abstract

Krüppel-like factor 1 (KLF1) plays a crucial role in erythropoiesis. In-depth studies conducted on mice and humans have highlighted its importance in erythroid lineage commitment, terminal erythropoiesis progression and the switching of globin genes from γ to β. The role of KLF1 in haemoglobin switching is exerted by the direct activation of β-globin gene and by the silencing of γ-globin through activation of BCL11A, an important γ-globin gene repressor. The link between KLF1 and γ-globin silencing identifies this transcription factor as a possible therapeutic target for β-hemoglobinopathies. Moreover, several mutations have been identified in the human genes that are responsible for various benign phenotypes and erythroid disorders. The study of the phenotype associated with each mutation has greatly contributed to the current understanding of the complex role of KLF1 in erythropoiesis. This review will focus on some of the principal functions of KLF1 on erythroid cell commitment and differentiation, spanning from primitive to definitive erythropoiesis. The fundamental role of KLF1 in haemoglobin switching will be also highlighted. Finally, an overview of the principal human mutations and relative phenotypes and disorders will be described.

## 1. Introduction

Krüppel-like factor 1 (KLF1) is an erythroid-specific transcription factor that plays a crucial role in erythropoiesis. Isolated for the first time from mouse erythroleukemia cell line (MEL), it was originally named erythroid Krüppel-like factor (EKLF) [1]. When related Krüppel-like factors were subsequently identified, the nomenclature was changed to KLF1 to reflect the order of discovery. The human *KLF1* gene is located on chromosome 19p13.2, whilst the mouse *Klf1* gene is located on chromosome 8. Both human and mouse genes are contained within approximately 3 kb of genomic DNA and consist of three exons (encoding 362 and 358 amino acids in humans and mice, respectively) and two introns. The KLF1 protein is characterised by two main functional domains: two N-terminal transactivation domains (TAD1 and TAD2) that recruit transcriptional activators, and three zinc finger (ZF) DNA-binding domains (ZF1, ZF2 and ZF3) located at the C-terminus (Figure 1A). The three ZFs share >90% sequence similarity and are expected to bind the same target sequence as mouse Klf1. The rest of the protein is proline-rich and retains approximately 70% sequence similarity to the mouse gene. Although its function as an activator is predominant, KLF1 may act as a repressor in some contexts [2,3]. SUMOylation, phosphorylation, acetylation and ubiquitination occur on KLF1 protein, especially in TADs, directing KLF1 interaction with protein inhibitor of activated STAT (PIAS) family members and p300/CREB and the SWI/SNF-related chromatin remodelling complexes, among others. These interactions regulate important functions of KLF1, such as its commitment to the erythroid lineage, the repression of megakaryocyte differentiation and its involvement in chromatin-hub creation to enhance β-globin expression [4,5,6,7].

KLF1 is a master erythroid gene regulator whose relevance in erythroid development and haemoglobin switching became clear as soon as the first *Klf1* KO mouse models were characterised [8,9] (Table 1). Mice lacking *Klf1* die at approximately embryonic day 15 (E15) due to severe anaemia, attributed to the loss of β-globin expression in the first instance, suggesting a β-thalassaemic phenotype. However, by restoring globin unbalance through human γ-globin expression in transgenic mice, haemolysis was not corrected and survival was not prolonged, which suggests an essential role of *Klf1* in the expression of other genes required for definitive erythropoiesis [10]. Murine *Klf11* null embryo foetal livers are near normal in size and rich in erythroid precursors; however, erythroid maturation, particularly haemoglobin accumulation, is impaired [9]. Erythroid cell numbers in circulation are drastically reduced and a higher presence of nucleated erythroblasts is highlighted [9]. 

Indeed, further analyses described a more extended range of Klf1 action covering the erythroid lineage commitment, the switching of globin genes from γ to β and the activation or suppression of a number of erythroid-specific genes, involved in red blood cell metabolism and structure, and the formation of chromatin remodelling complexes [11]. 

The present review will focus on some of the principal tasks of KLF1 in erythropoiesis control and erythrocyte function. Its involvement in primitive and definitive erythropoiesis, in the cell cycle and in the function of the central macrophage of the erythropoietic island will be illustrated. Thereafter, KLF1 control of haemoglobin switching and the regulation of KLF1 expression by non-coding RNA will be discussed. Finally, an overview of the principal human KLF1 variants and relative phenotypes and disorders will be reported.

**Table 1 cells-11-03069-t001:** List of KLF1-affected genes cited in the present review.

Gene Symbol	Name/Function	References
**CELL-CYCLE REGULATORS**		
E2F2	E2F Transcription Factor 2	[12,13,14]
P18 (CDKN2C)	Cyclin Dependent Kinase Inhibitor 2C	[14,15]
P21(CDKN1A)	Cyclin Dependent Kinase Inhibitor 1A	[14,15]
P27 (CDKN1B)	Cyclin Dependent Kinase Inhibitor 1B	[14,15]
**HAEMOGLOBIN REGULATION**		
HBB	Human adult haemoglobin subunit β	[8,9,16]
Hbb-b1/Hbb-b2	Murine adult haemoglobin subunit β	[1,8,9,16]
HBG	Human foetal haemoglobin subunit γ	[16,17]
Hbb-y	Murine embryonic haemoglobin subunit εy	[2,18,19]
Hbb-bh1	Murine embryonic haemoglobin subunit βh1	[2,18,19]
Hba-x	Murine embryonic haemoglobin subunit ζ	[2,18,19]
BCL11A	B cell CLL/lymphoma 11A	[20,21,22,23,24,25]
ZBTB7A	Zinc finger and BTB domain containing 7A	[26]
**ADHESION MOLECULES/ANTIGENS**		
BCAM	Basal cell adhesion molecule (Lutheran blood group)	[27]
VCAM	Vascular cell adhesion molecule	[28]
CD44	CD44 molecule (Indian blood group)	[29,30,31]
P1PK	alpha 1,4-galactosyltransferase (P blood group)	[2,32,33]
LW (ICAM4)	Intercellular adhesion molecule 4 (Landsteiner–Wiener blood group)	[2,32,33]
KNOPS	Complement C3b/C4b receptor 1 (Knops blood group)	[2,32,33]
OK	Basigin (Ok blood group)	[2,32,33]
RAPH	CD151 molecule (Raph blood group)	[2,32,33]
ERMAP/SCIANNA	Erythroblast membrane-associated protein (Scianna blood group)	[2,32,33]
AQP1	Aquaporin 1 (Colton blood group)	[29]
**HEME SYNTHESIS/IRON PROCESSING**		
ALAS2	5’-aminolevulinate synthase 2	[34]
ALAD	Aminolevulinate dehydratase	[34]
HMBS	Hydroxymethylbilane synthase	[34]
SLC25A37	Solute carrier family 25 member 37	[34,35]
STEAP3	STEAP3 metalloreductase	[34,35]
ABCG2	ATP binding cassette subfamily G member 2 (Junior blood group)	[34,35]
ABCB10	ATP binding cassette subfamily B member 1	[34,35]
TFR2	Transferrin receptor 2	[34,35]
**OTHER CATEGORIES/FUNCTIONS**		
PKLR	Pyruvate kinase L/R	[36]
DNASEII-ALPHA	Deoxyribonuclease 2, lysosomal	[37]
TER119	Lymphocyte antigen 76	[2]
AHSP	Alpha haemoglobin stabilizing protein	[2,38,39]
CD9	CD9 molecule	[2,18]
CD24	CD24 antigen (small cell lung carcinoma cluster 4 antigen)	[2,18]
DEMATIN	Erythrocyte membrane protein Band 4.9	[2,40]
MGST3	Microsomal glutathione S-transferase 3	[2,18]
ACP3	Acid phosphatase 3	[2,18]
BZRP (TSPO)	Translocator protein	[2,18]
RH-CDE	Rhesus CDe complex	[2,18]

## 2. Erythropoiesis

During haematopoiesis, Klf1 expression pattern has been well described in mice and highlights its specificity for the erythroid lineage (Figure 1B). Klf1 is already present, albeit at low levels, in haematopoietic stem cells (HSC), multipotent progenitors (MPP), and common myeloid progenitors (CMP) when separation from the common lymphoid progenitors (CLP) occurs; however, it tends to increase in the subsequent steps of commitment, thereby driving erythroid differentiation. In particular, it increases from CMP to megakaryocyte/erythrocyte progenitors (MEP) when separation from the granulocyte-monocyte progenitors (GMP) occurs, and from MEP to the erythroid progenitors, burst-forming unit-erythroid (BFU-E) cells and colony-forming unit-erythroid (CFU-E) cells, that give rise to the terminal phase of erythropoiesis. In this final step, where the fate of the lineage has been determined, Klf1 expression is markedly increased (Figure 1B) in erythroid cells, whilst it remains expressed at low levels in megakaryocytes [41,42]. Erythroid expression of KLF1 is mediated by GATA1 binding to its promoter [43] and KLF1 and GATA1 act together to regulate the expression of many erythroid genes [35]. The process of erythropoiesis occurs in different organs during mammalian embryonic development. Primitive erythropoiesis occurs in the yolk sac, in which primitive erythroid precursors (i.e., erythroid-colony-forming cells, Ery-CFCs) arise and further differentiate in the bloodstream [44,45,46]. Definitive erythropoiesis occurs in the foetal liver until birth and subsequently in the bone marrow (Figure 2) [46]. Definitive erythroblasts mature in specialised niches called erythroblastic islands, which are characterised by the central macrophages of erythroid islands (CMEIs) surrounded by maturing erythroblasts, which play a fundamental role in erythropoiesis homeostasis and nuclei digestion (Figure 1D) [47,48]. Both Ery-CFC and definitive erythroid progenitors (BFU-E/CFU-E), undergo a further phase of differentiation, called terminal erythropoiesis (Figure 1C) starting from large nucleated cells, proerythroblasts, that differentiate to basophil, then polychromatic and, finally, orthochromatic erythroblasts. Each phase of differentiation is determined by one or two mitosis events and a progressive decrease in cell size, increase in haemoglobin production and the progressive DNA condensation of erythroid cells. The extrusion of condensed nuclei occurs in definitive orthochromatic erythroblasts that become reticulocytes and are released in the bloodstream, where haemoglobin levels increase and reticulocytes differentiate in red blood cells (RBC) (Figure 1C) [48]. The regulation of events that accompany the entire erythropoietic process is of the utmost importance for the homeostasis of the entire organism. Notably, transcription factors such as Klf1 serve a pivotal role.

### 2.1. Role of Klf1 in Primitive Erythropoiesis 

KLF1 expression in mice begins as early as the first primitive erythroblasts can be identified in the yolk sac, at approximately embryonic day 7.5 (E7.5) and persists in developing primitive erythroblasts in peripheral blood [18,49]. Despite this, Klf1 activity in primitive erythropoiesis was elusive, since mice lacking Klf1 expression manifest barely noticeable alterations in both morphology and functionality of erythroblasts until definitive erythropoiesis occurs [8,9]. Generally, the set of genes regulated by Klf1 in primitive erythropoiesis does not seem to show new elements when compared to its activity in definitive erythropoiesis. On the other hand, and unlike in definitive erythropoiesis, compensation for Klf1 deficiency by other transcription factors—and particularly by other members of the Krüppel-like factor family (e.g., Klf2, also called Lklf)—occurs in primitive erythropoiesis [19]. Klf2 is an important regulator of haematopoiesis and embryonic globin expression [50]. Klf1/ Klf2 double-knockout mice exhibit a more severe phenotype than what is observed in single KO mice for each Krüppel-like gene and die earlier at E11.5 [50,51,52]. A regulatory network involving Klf1/ Klf2 cooperation has been identified in the activation of *MYC* gene expression (*c-MYC*) in mice, [53] a gene involved in primitive erythroblast development starting from the proerythroblast stage.

Expression-profile studies on foetal livers from *Klf1*^+/+^ and *Klf1*^-/-^ mice have accounted for a broad range of *Klf1* target genes. Through RT-PCR and Western blotting analyses, a number of these genes have also been found to be targets of Klf1 in the embryo [2]. Among them, Ter119, a protein strongly associated with glycophorin A and a key marker in the analysis of erythropoiesis, was highlighted [2,54,55]. 

Another highly down-regulated gene in Klf1^-/-^ mice embryos is *Dematin*, an essential component of the cytoskeleton in erythroblasts [40] that is also regulated by Klf1 in definitive erythropoiesis. *Dematin* absence leads to an increase in cellular membrane fragility, which could explain the morphological abnormalities of the cell membrane described in *Klf1*-null mice embryos [2,40]. The list of genes affected by a lack of Klf1 in mice embryos also includes *Ahsp*, *Mgst3*, *Ermap/Scianna*, *Acp3*, *Bzrp*, *Rh-cde*, *Icam4/Lw*, *Cd9* and *Cd24* [2,18]. The effects of Klf1 on embryonic globin expression went unnoticed until 2007 [19] probably due to the fact that the effect of Klf1 deprivation on embryonic globin gene expression is mild compared to that on definitive erythropoiesis globin genes. Despite this, the presence of Heinz bodies—which are signs of α/β-like globin imbalance—was previously observed in primitive red cells from *Klf1*^-/-^ mice [2,56]. RNA-Seq analysis of *Klf1^-/-^* mice revealed a twofold down-regulation of εy- and βh1-globin expression with respect to *Klf1*^+/+^ mice and threefold lower levels of ζ-globin [19]. *Klf1* and *Klf2* double-KO mice revealed the additional down-regulation of embryonic globins as a further demonstration of the co-operational regulation of the two transcription factors in primitive erythropoiesis [19].

### 2.2. Role of Klf1 in Definitive Erythropoiesis 

As described in Section 2.1, the impact of Klf1 in definitive erythropoiesis is certainly more pronounced than what is observed in primary erythropoiesis. For this reason, the role of KLF1 in definitive erythropoiesis has been characterised in more detail. Several studies have explored its implication in different aspects of erythroid differentiation, such as cell-cycle progression, expression in CMEIs and erythroid lineage commitment.

#### 2.2.1. Role of Klf1 in Cell-Cycle Regulation

The cell cycle is a key mechanism in the maturational process of erythroblast cells (Figure 1C and Figure 1D) and its fine regulation is required to achieve the optimal production of RBCs and ensure oxygenation of all tissues. In this contest, Klf1 serves a crucial role in both the activation of proliferative and anti-proliferative genes expression [12,13]. Definitive erythropoiesis in mice lacking *Klf1* expression shows an accumulation of erythroid progenitors at the early stages of maturation (BFU-E and CFU-E) and a failure in the proliferation of more mature erythroblast cells [12,13]. The ultimate effect of this perturbation is the exasperation of anaemia in the mouse model. Expression profiling on E13.5 foetal livers from *Klf1*^+/+^ and *Klf1*^-/-^ mice revealed several down-regulated genes that are involved in cell-cycle regulation. Among these, particularly important is the role played by the *E2f2* gene, which is directly targeted by Klf1 [12,13,14]. E2f2 regulation of the cell cycle is connected to the phosphorylation levels of the retinoblastoma (RB) complex, which depends on cyclins/cyclin-dependent kinase activation [57]. Activated E2f2, following RB protein phosphorylation, triggers DNA synthesis and an exit from the G_1_ state [12,13]. In mice lacking Klf1 expression, blocking the cell cycle consequent to the down-regulation of proliferative proteins does not seem to affect all erythroid progenitor cells. Those erythroblasts that escape cell-cycle blockage can differentiate further and mature until the later stage of orthochromatic erythroblasts, even if at a reduced number [15]. Klf1 control of the cell cycle in erythroid cells has been highlighted not only in genes that regulate the G_1_-S phase but also in genes that inhibit cell cycles, such as *p18, p21* and *p27* [14,15]. The effects of Klf1 on cell-cycle inhibitors do not appear to perturb the maturational development of erythroblasts until the orthochromatic stage. The maturational fate of orthochromatic cells involves the extrusion of the nucleus, with cell-cycle blockage being required for this to occur. Defective cell-cycle exit, due to the down-regulation of inhibitors, leads to a failure to enucleate. The persistence of the S phase has been also observed in vitro in *Klf1*^-/-^ ESRE (extensively self-renewing erythroblasts) human cells [15]. As will be discussed in the next section, the enucleation process in mice lacking Klf1 expression is further impaired by a non-cell autonomous process involving the central macrophage of erythroblastic islands [58].

#### 2.2.2. Role of Klf1 in Central Macrophages of Erythroblastic Islands (CMEI) functions

The expression of Klf1 on macrophages was first highlighted in human primary cells, and its involvement in the regulation of IL12-p40 expression was also demonstrated [59]. Subsequently, a further demonstration of both the expression and activity of Klf1 in central macrophages of erythroblastic islands was described in vivo [28,37]. Recently, a Klf1+ F4/80+ macrophage population in murine foetal liver cells with a peculiar pattern of expression was identified. This population presents both erythroid and splenic macrophage characteristics but also expresses a unique group of genes [60]. Notably, a principal component analysis (PCA) of sorted Klf1+ F4/80+ and Klf1- F4/80+ from murine foetal liver indicated the enrichment of genes involved in erythroid functions (i.e., heme synthesis, iron transportation and myeloid/erythroid differentiation) in Klf1+-expressing macrophages, whilst Klf1- macrophages presented a profile mostly connected to innate and cellular immunity [60]. Interestingly, erythropoietin receptor (EpoR) was one of the enriched genes in Klf1+ foetal liver macrophages. The presence of EpoR on erythroblastic island macrophages has previously been identified as a distinctive tract of erythroid island macrophages in bone marrow [61]. Moreover, the induction of KLF1 expression in human induced pluripotent stem-cell-derived macrophages (iPSC-derived macrophages) is able to lead their differentiation into a CMEI phenotype as shown by the up-regulation of genes related to CMEI function. Notably, these erythroid-island-like macrophages were able to establish an erythroid island niche in vitro and to lead erythroblast differentiation until full maturation, as demonstrated by the augmented absolute number of mature enucleated erythroid cells compared with controls [62]. Adhesion molecules play a fundamental role in erythroid differentiation. The optimal development of erythropoiesis requires direct erythroid–erythroid and macrophage–erythroid interaction (Figure 1D) [48]. Erythroid–macrophage interaction is perturbed in *Klf1* KO mice due to the down-regulation of two adhesive systems. The lack of *Klf1* results in down-regulation of Icam4 adhesion molecules in erythroid cells [2,3,12,18], whilst the Vcam1 adhesion molecule is affected by the absence of *Klf1* in central macrophages (Figure 1D) [28]. Vcam1 and Icam4 interacting receptors, α4β1-integrin expressed on erythroblasts and αv-integrin present on macrophage surfaces were not found to be Klf1 targets (Figure 1D). However, the down-regulation of Vcam1 and Icam4 on the two erythropoietic cellular elements disrupts island integrity and thereby contributes to the dysregulation of the entire erythroid maturational process. The absence of Klf1 affects another essential role of macrophages in the erythroid niches, which consists of the engulfment of extruded nuclei and their subsequent degradation in lysosomes (Figure 1D). Fundamental to this function is the expression of DnaseII-alpha protein in macrophages, which hydrolyses nucleic DNA and causes its degradation [37,63]. Klf1 expressed in CMEIs binds and activates the promoter of DnaseII-alpha, which is severely down-regulated in the *Klf1* KO mouse foetal liver [37]. Accumulation of extruded nuclei in CMEI lysosomes, due to the inability to degrade nucleic DNA, impairs CMEI capacity to sustain erythropoiesis and triggers the production of anti-proliferative cytokines such as interferon-β, which has cytotoxic effects and inhibits erythroid maturation [37,64]. Moreover, the block of erythroid maturation also affects their capacity to extrude their nuclei so that nucleated definitive red cells are present in blood and the number and function of erythroblastic islands in the foetal liver are impaired. As a demonstration of this mechanism, the crossing of *Klf1*^-/-^ mice with mice in which the receptor for interferon-β was deleted (*Ifnar*^-/-^ mice), so that the response to the cytokine was abrogated, resulted in a partial rescue of *Klf1*^-/-^ mice with a delay of embryo death until E16.5 [58], the amelioration of erythropoietic kinetics and the presence of enucleated reticulocytes in foetal liver at E14.5 [58]. 

#### 2.2.3. Role of KLF1 in Erythroid Commitment

An important role of Klf1 in megakaryocyte/erythrocyte balance has been suggested by the observation that Klf1 gain- or loss-of-function experiments in mouse models are characterised by an altered balance between platelets and RBC count [42,65]. Observations from mouse models highlight not only an increase in platelet count after the abrogation of Klf1 expression but also the persistence of megakaryocytic surface markers in differentiating erythroblast cells [2,66]. These data indicate that Klf1 is involved in erythroid lineage commitment. Functional cross-antagonism between KLF1 and friend leukaemia integrator 1 (Fli-1), a key activator of the megakaryocytic lineage, acts in delineating the balance between the two choices that MEPs face during their commitment [42,67]. The silencing of Klf1 in human erythroid cells increases Fli-1 occupancy at promoters and enhancers of genes that promote megakaryocytic differentiation [68]. At the same time Fli-1 is able to repress Klf1 stimulation of erythroid-specific genes, such as β-globin. [67]. A possible explanation for this mutual repression resides in their rivalry in protein–protein binding with Gata-1 protein, an important co-regulator of both erythroid and megakaryocytic lineages [68]. An additional player in Klf1/Fli-1 antagonism is runt-related transcription factor 1 (Runx1), protein which promotes megakaryocyte differentiation by repressing Klf1 expression in committed megakaryocytes, thus promoting Fli-1 activity [69]. 

## 3. Haemoglobin Switching

The mouse β-globin cluster, localised on chromosome 7, presents four genes and undergoes only one switching event from embryo to foetal-adult. Embryonic globins are represented by εγ- and βh1-globins, which are expressed in the yolk sac from E7.5, whilst adult globins β^min^ and β^maj^ are expressed in foetal livers starting from E11.5 and become the only detectable globins in circulation at approximately E16.5 (Figure 2). Human β-globin cluster is localised in chromosome 11 and consists of five genes: ε-globin expressed in the yolk sac for the first 3 months of intrauterine life; Aγ and Gγ globin, which are expressed in the foetal liver from month 3 until birth; and β-globin and δ-globin, which are expressed in the bone marrow soon after birth (Figure 2). Therefore, throughout the human lifespan, there are two switching events: the first from embryonic to foetal haemoglobin around the third month of gestation, and the second from foetal to adult haemoglobin starting from birth (Figure 2) [70,71,72,73,74,75].

Although Klf1 involvement in β-globin expression has been hypothesised since its first description [1], its role became clear as soon as *Klf1^-/-^* mice were obtained to characterise the gene functions [8,9,16] Moreover, transgenic mice containing the full human β-globin locus crossed with *Klf1*-deficient mice showed that the silencing of β-globin expression consequent to *Klf1* deprivation is accompanied by increased levels of foetal haemoglobin, whilst a delay in the γ- to β-globin switch was observed in *Klf1*^+/-^ mice [17,76]. The switching of haematopoiesis from the yolk sac to the foetal liver in mice results in a threefold increase in Klf1levels [16]. These data identify Klf1 as a key player in foetal to adult haemoglobin switching. The activation of β-globin is a direct function of Klf1, which binds to the CACCC box present on the *Hbb* gene promoter as well as on the β-globin locus control region (LCR) hypersensitive sites (HS), thereby allowing the formation of the loop LCR-*Hbb* gene and triggering β-globin production [1,70,77,78]. Moreover, the disruption of the Klf1 binding site on the proximal β-globin promoter through clustered regularly interspaced short palindromic repeat (CRISPR) mediated gene editing is sufficient to disable the normal competition for the LCR and reactivate foetal haemoglobin [79]. On the other hand, the repression of γ-globin expression is not a direct function of KLF1, which exerts this role by activating BCL11A, which subsequently represses HBG expression. The discovery of the role of BCL11A in haemoglobin switching occurred due to genome-wide association studies (GWAS) [20,21] that associated the gene with HbF production and the persistence of foetal haemoglobin in humans affected by β-thalassemia intermedia and sickle cell disease [21,22,23]. In BCL11A^-/-^ mouse models carrying the human β-globin locus, the γ-globin gene is not silenced in definitive erythropoiesis. Furthermore, heterozygosity for the BCL11A null gene can also decrease γ-globin repression [24].

Compound KLF1:BCL11A mutant mice also demonstrate the KLF1/BCL11A interplay for the completion of switching. Once erythropoiesis starts in the foetal liver, KLF1 levels increase, thereby triggering both β-globin and BCL11A expression. Increased levels of BCL11A then block γ-globin transcription. Finally, the blockage of γ-globin transcription removes a spatial obstruction on the β-locus, thereby allowing more KLF1 proteins to bind to the β-globin promoter [25].

More recently it has been shown that KLF1 also directly drives the expression of ZBTB7A, a further γ-globin repressor, in erythroid cells by binding to its proximal promoter [26].

## 4. Regulation of KLF1 Expression by Non-Coding RNA

The epigenetic regulation of KLF1 by non-coding RNA (ncRNA) has been investigated in the last few years. In particular, microRNAs (miRNAs or miRs), a class of small, non-coding linear RNA, represent an interesting field of investigation due to the relevance of this class of molecules as a therapeutic target [80]

MiR-326 has been identified as a direct inhibitor of KLF1 and positive correlation has been evidenced between its expression and foetal haemoglobin levels in reticulocytes from β-thalassemia major patients compared with healthy controls [81]. Moreover, altered KLF1 expression has been highlighted in K562 cells stably expressing MiR-34a [82] and the correlation between KLF1 levels and the up-regulation of miR-451 has been reported in murine embryonic stem cells [83]. The down-regulation of KLF1, BCL11A and MYB associated with up-regulation of miR-15a, miR-16, miR-26b and miR-151-3p has been described in K562 and differentiated erythroid human primary cells treated with hydroxyurea, suggesting a possible association between these miRNAs and the critical regulators of HbF expression [84]. 

Long non-coding RNAs (lncRNAs) are implicated in the regulation of KLF1 as well. Recently, a deep RNA-sequencing study identified a novel lncRNA (Gm15915), named lncEry, for its elevated expression in erythroid cells. lncEry seems to promote KLF1 expression, through interaction with WD repeat-containing protein 82 (WDR82) affecting early and late stages of erythropoiesis [85].

## 5. Human KLF1 Variants and Phenotypes 

According to the Human Gene Mutation Database (https://www.hgmd.cf.ac.uk/ac/index.php, accessed on 31 May 2022), more than 130 KLF1 variants have been reported to date. These are classified into five classes corresponding to their effect on the gene: FP, functional polymorphism; DM, disease-causing mutation; DM? supposed disease-causing mutation; DP, disease-associated polymorphism; and DFP, disease-associated polymorphism with supporting functional evidence. The variants described to date have mostly been localised in the coding region, causing alterations in the primary structure of the protein. A smaller percentage (2.3%) are located in the *KLF1* promoter, affecting gene transcription. Mutations in the coding sequence can be further grouped into four subclasses according to their effect on protein function [86]: class 1 is represented by missense variants located outside the ZF domains, with no or minor functional effects; class 2 includes missense variants or small in-frame deletions that interfere with the normal function of KLF1, mostly located in the ZF domains; class 3 comprises stop codon or frameshift variants that result in truncated KLF1 proteins lacking the ZF domains; in class 4, a unique mutation (p.Glu325Lys) is localised in a highly conserved residue in ZF2, causing dominant severe congenital dyserythropoietic anaemia type IV (see Section 5.3) [29,87]. KLF1 regulates approximately 700 genes in human erythroid cells, which are involved in a wide variety of molecular processes. 

Many of them have previously been reported as murine KLF1 target genes [3]. These genes are sensitive not only to KLF1 expression levels (i.e., when one allele carries an inactivating mutation) but also to the type of *KLF1* mutation. Indeed, specific *KLF1* variants result in the altered expression of these genes, causing a wide range of benign and severe erythroid phenotypes. This emphasises the pleiotropic effect of KLF1 in wild-type and mutant conditions (Table 2). These phenotypes include the In(Lu) blood types, HPFH, increased levels of HbA_2_ (α_2_δ_2_), congenital dyserythropoietic anaemia, hydrops foetalis, non-spherocytic haemolytic anaemia, red cell protoporphyrin, and pyruvate kinase deficiency (Table 2).

### 5.1. In(Lu) Phenotype 

The Lutheran blood group system consists of 29 antigens encoded by the red cell membrane glycoprotein basal cell adhesion molecule (BCAM). Since the expression of BCAM is highly sensitive to the level of functional *KLF1*, its variants cause a marked down-regulation of the *BCAM* gene generating the In(Lu) (inhibitor of Lutheran blood group) phenotype, as demonstrated by a transcriptomic analysis of erythroblasts from In(Lu) type donors when compared to a control group [27]. Further evidence of the impact of human *KLF1* variants on the Lutheran (LU) system was provided by a sequencing analysis performed on the genomic DNA of In(Lu) donors. The analysis showed mutations in the *KLF1* promoter or coding sequences in 21 out of 24 subjects. All cases were in the heterozygous state [27]. Nine different mutations were detected (Table 2). Six of the nine mutations occurred in the *KLF1* DNA ZF domains. One of these (Arg319GlufsX34) is predicted to cause a frameshift and another (Lys292Term) encodes a premature termination codon, whilst all others result in single amino acid substitutions (His299Tyr, Arg328Leu, Arg328His and Arg331Gly). The three mutations outside of the ZF domains result in a premature termination codon (Leu127Term), a frameshift (Pro190LeufsX47) and an altered GATA1 binding site in the *EKLF* promoter (-124T>C) [27]. Since the discovery of the effect of KLF1 on blood-group phenotypes, genotyping of the *KLF1* variant has become the focus of In(Lu) phenotype research. Unfortunately, very few population-based studies have been performed in this field [32,96,97], despite being particularly important in the transfusion management of difficult transfusion-related cases. *KLF1* variants are also associated with the reduced expression of other red cell membrane proteins that carry blood group antigens, such as the Indian (IN), P1PK, Landsteiner–Wiener (LW), Knops (KN), OK, RAPH and I blood group systems; however, most are less sensitive to its levels [2,12,13,27,32,33,98,99,100].

### 5.2. Globin-Expression Dysregulation 

KLF1 transcription factor regulates the β-globin gene expression by binding to the CACCC box of the β-globin promoter [1]. A variant in this region (-87 mutation) causes β^+^-thalassemia with elevated HbF levels [101,102]. *KLF1* is a master erythroid gene activator, as demonstrated by the fact that the -198 mutation localised in the γ-promoter creates a new binding site for *KLF1* that causes the British hereditary persistence of the HbF (HPFH) phenotype [103]. HPFH was also associated to the haploinsufficiency of KLF1, which was described for the first time in a Maltese family with HPFH (HbF levels: 3.3–19.5%) and caused by a nonsense (Lys288Term) mutation on a *KLF1* allele (Table 2). In the same family, functional studies revealed low levels of BCL11A, implying that the main cause of HPFH is an impaired expression of BCL11A, which is sensitive to KLF1 dosage [88]. However, subsequent data from a Sardinian family with HPFH caused by compound heterozygosity for two *KLF1* mutations did not support the haploinsufficiency of KLF1 as a cause of HPFH [90]. *KLF1* gene sequencing performed on two brothers of this family, with a marked increase in HbF (22.1–30.9%), revealed a genetic compound condition for a nonsense mutation (Ser270Term) inherited from the father and a missense mutation (Lys332Gln), inherited from the mother. The Ser270Term nonsense mutation is predicted to completely ablate the ZF domain and the ability of KLF1 to interact with DNA. The missense Lys332Gln mutation lies in the second KLF1 ZF domain and combination with the Ser270Term nonsense mutation further reduces KLF1 function (Table 2). In this family only individuals with two in trans *KLF1* mutations have HPFH, while the monoallelic loss of *KLF1* expression is associated with normal HbF levels. Furthermore, in this family, very high levels of zinc protoporphyrin associated with *KLF1* mutations were reported (discussed in Section 5.6) [90].

Finally, *KLF1* variants are associated with moderately increased levels of HbA2 (Table 2). The role of *KLF1* variations leading to a borderline HbA2 phenotype was first reported in the Sardinian population [89]. Overall, 52 of 145 subjects (35.9%) with borderline HbA2 were heterozygotes for *KLF1* mutations. Among these, the nonsense Ser270Term was the most frequent (80.8%), being found in 42 individuals. The mean HbA2 levels in these patients were 3.6% ± 0.2 (reference value: 2.8% ± 0.2), with normal MCV and MCH [89]. Another study involving 65 borderline HbA2 cases (HbA2: 3.3–3.9%) revealed that 7.6% of these carried *KLF1* variations. The mean HbA2 level of these cases was found to be 3.52% ± 0.14. One patient homozygous for Phe182Leu variation with HbA2 of 3.4%, and HbF of 2.5%, was also found [30]. The molecular mechanism for borderline HbA2 is most likely explained by an increased LCR interaction with the competing *HBD* gene when compared to the *HBB* gene under diminished KLF1 expression, which increases the δ-globin gene expression with a consequent increase in HbA2 level [30].

### 5.3. Congenital Dyserythropoietic Anaemia (CDA) 

CDA is a rare inherited red blood cell disorder with hallmarks of morphologic abnormalities of erythroblasts in the bone marrow and ineffective erythropoiesis and haemolysis, which can be divided into four subtypes (I–IV) based on erythroblast nuclear morphology [104]. The molecular aetiology of most CDA subtypes has been solved: type I is caused by mutations in codanin-1 (CDN1); type II is caused by mutations in SEC23B; type III is caused by mutations in KIF23 [105,106,107]. Rare type IV CDA cases are caused by a dominant mutation in *KLF1* (Table 2) [29,31,87,108]. The mutation consists of a G-to-A transition in one allele of *KLF1* exon 3, a change that results in the substitution of glutamic acid 325 by a lysine (p.Glu325Lys) in ZF2, which is highly conserved across KLF proteins from different species and across the entire KLF family. The p.Glu325Lys mutation leads to the abolition of the expression of the water channel AQP1 and the adhesion molecule CD44, and a reduced expression of two other adhesion molecules (BCAM and ICAM4). The peripheral blood contains evidence of poikilocytosis, anisocytosis, fragmented erythrocytes and many nucleated red blood cells. These nucleated red blood cells are mostly orthochromatic erythroblasts, which suggests a failure of terminal erythroid differentiation. Additionally, CDA IV is characterised by the very high expression of HbF (∼35% of total Hb) and persistent expression of embryonic globin (HB-Portland) [29]. Notably, a variant in the homologous residue in mouse *Klf1*, Glu339Asp, underlies semi-dominant neonatal anaemia (Nan) [109,110]. Phenotypically, Nan mice display many similarities with CDA IV patients, and a limited subset of Klf1target genes is down-regulated in Nan mice [109,111,112], which may be helpful for the analysis of CDA IV patients. The E325K mutation reveals an altered DNA-binding specificity, both at canonical target sites (i.e., β-globin promoter) and aberrant binding to new sites causing ectopic transcriptional activation of non-erythroid genes. Based on in vivo and in vitro experiments, it is likely that CDA IV is caused by a general dysregulation of gene expression in developing erythroblasts that interferes with differentiation causing haemolysis. The E325K allele is probably hypomorphic to some known KLF1 target genes, such as CD44 and ICAM4, and neomorphic to other nonerythroid genes [112,113,114,115].

### 5.4. Hydrops Foetalis

The first case of a KLF1-null human was described in a patient with hydrops foetalis caused by compound heterozygosity for the Trp30Term and Arg319GlufsX34 null alleles (Table 2) [34]. The proband was born with severe non-spherocytic haemolytic anaemia (NSHA), hepatosplenomegaly and jaundice, which was difficult to control. There was marked erythroblastosis with erythroid expansion within the bone marrow and marked (76%) HPFH. The proband underwent a successful unrelated allogeneic bone marrow transplantation at 6 years of age but developed cerebral palsy due to kernicterus [34]. RNA-Seq was performed on peripheral circulating blood cells of the proband and his parents and compared with published RNA-Seq datasets. Notably, 819 erythroid genes were poorly expressed when compared to normal definitive erythroblasts. Many of these have been previously reported as murine Klf1 target genes [2,3,99,116]. These encode for cytoskeletal proteins, heme synthesis enzymes, cell-cycle regulators, blood-group antigens and the chaperone AHSP (alpha-haemoglobin-stabilising protein). AHSP prevents the aggregation of α-globins during haemoglobin assembly [38]. It is a direct target gene of KLF1 and loss of expression in KLF1 mutant cells worsens the damage conferred by excess α-globin chains in β-thalassemia [39,117,118]. 

Moreover, the pattern of altered gene expression in erythroid cells was very similar to that reported for *Klf1*-null mice [2,12,34]. It is unclear whether the persistence of HbF explained birth survival or whether this infant had complementary variants that ameliorated the effects of KLF1 deficiency. Another report confirmed that in humans, although compatible with life, the loss of KLF1 severely impairs erythropoiesis [119].

### 5.5. Non-Spherocytic Haemolytic Anaemia (NSHA)

NSHA is a term for hereditary anaemias characterised by the reduced survival of red blood cells with abnormal morphology, erythroid hyperplasia in the marrow, and haemolysis. The term is usually applied once thalassemia, hereditary spherocytosis, enzymopathies and CDA have been excluded by molecular, enzymatic and morphological tests. The patients described to date have had transfusion-dependent haemolytic anaemia originally misdiagnosed as thalassaemia and are compound heterozygous for *KLF1* variants: Ala298Pro/Gly176AlafsX179 [91], Pro338Ser/Gly176ArgfsX179 [92], Pro338Thr/Gly176ArgfsX179 [93], Gly335Arg/Gly176ArgfsX179 [94], Arg331Trp/Gly335Arg, Arg301His/Gly176ArgfsX179, -154 C>T/Ala298Pro, Gln58Ter/Ala298Pro, Ala298Pro/Gly176ArgfsX179 [95]. Patients who are compound heterozygous for the latter two mutations also showed reduced levels of pyruvate kinase enzyme activity (Table 2; see Section 5.7) [95]. 

### 5.6. Red Cell Protoporphyrin

A family of Sardinian origin (described in Section 5.2 for HPFH) has been reported showing very high levels of zinc protoporphyrin in red blood cells (detected value of 306 μg/dL; reference value < 35 μg/dL). This phenotype was associated, as for HPFH, with compound heterozygous mutations of *KLF1* (Table 2) [90]. In patients with *KLF1* variants, iron levels are normal but iron is not efficiently incorporated into heme. In this situation, zinc may be incorporated into the heme instead of iron.

KLF1 coordinates expression of many of the genes involved in iron metabolism of erythroid precursors, including heme synthesis enzymes (e.g., ALAS2, ALAD, HMBS) [3,12,56,120] and proteins regulating the processing of iron (e.g., TFR2, SLC25A37, STEAP3, ABCG2, and ABCB10) [34,35]. KLF1 role in iron metabolism was demonstrated in a Nan mutant mouse that manifested a dramatic increase in the zinc protoporphyrin. This mouse model carried the Glu339Asp missense mutation, localised in the central Zn finger, which alters the DNA binding domain. The failure of *KLF1*-deficient cells to accumulate haemoglobin could be partly due to reduced heme synthesis since both processes are tightly linked [115,121]. 

### 5.7. Pyruvate Kinase Deficiency

Mutations in the *KLF1* gene in Thai paediatric patients have been described to cause the severe down-regulation (50% of normal) of pyruvate kinase (PK) levels, resulting in NSHA [95]. The PK enzyme is encoded by the *PKLR* gene and, once causative mutations in the coding region of *PKLR* were excluded by DNA sequencing, the *KLF1* gene was analysed, since it binds the *PKLR* gene promoter [36]. As stated in Section 5.5, all patients with the NSHA phenotype and PK deficiency resulted in compound heterozygotes for the same *KLF1* gene mutations (Ala298Pro/Gly176ArgfsX179) (Table 2) [95]. 

Patients’ PK levels were pathologically reduced and abnormal red blood cells resembling the typical punctiform cells of PK deficiency were observed. 

## 6. Conclusions and Future Perspectives 

Understanding the complex network of mechanisms governing erythropoiesis and haemoglobin switching is of primary importance in haematology. In vivo functional studies on animal models and in vitro studies on human erythroid cells have made it possible to unravel the central role of KLF1 in these processes [1,2,3,8,9,15,34].

Of particular importance is its involvement in γ- to β-haemoglobin switching due to its beneficial impact on disorders, such as β-thalassemia and sickle cell anaemia [8,9,17,103]. Clinical evidence indicates that the re-activation of foetal γ-globin gene expression can ameliorate the clinical course of β-hemoglobinopathies. The finding that KLF1 regulates β-globin gene expression and that reduced levels of KLF1 expression cause high levels of foetal haemoglobin would have identified it as a good therapeutic target. However, the modulation of KLF1 is difficult to contemplate as a therapeutic strategy in clinical practice due to its universal role in red blood cell homeostasis. KLF1 is a master activator of erythroid-specific gene expression as also evidenced by the fact that naturally occurring mutation creating a de novo binding site for KLF1 in the γ-globin gene promoter led to HPFH. This observation justifies an alternative strategy for using KLF1 as a therapeutic tool for β-hemoglobinopathies. Introducing natural KLF1-binding HPFH mutations via gene editing-based strategies, such as CRISPR-Cas9 DNA-base-editing technologies, could represent a possible approach, since the clinical consequences of silencing crucial genes for red blood cell homeostasis would be overcome [103,122].

The development of new molecular technologies (i.e., GWAS, NGS, ChIP-Seq, transcriptome, exome, etc.) has facilitated the identification of many variants in the *KLF1* gene linked to RBC phenotypes and disorders. This has provided a better understanding of the function of KLF1. Moreover, genotype–phenotype correlation evidenced that in cases of increased RBC and HbF, as well as borderline increases in HbA2 and ZnPP with normal iron stores, further molecular investigations should be performed to understand the KLF1-carrier status of patients [27,29,30,34,55,56,57,88,90,91,94,95,96,99]. Additionally, these findings have made it possible to identify *KLF1* as an important gene disease, which explains a significant proportion of unexplained RBC disorders and may lead to the development of new therapeutic strategies.

## Figures and Tables

**Figure 1 cells-11-03069-f001:**
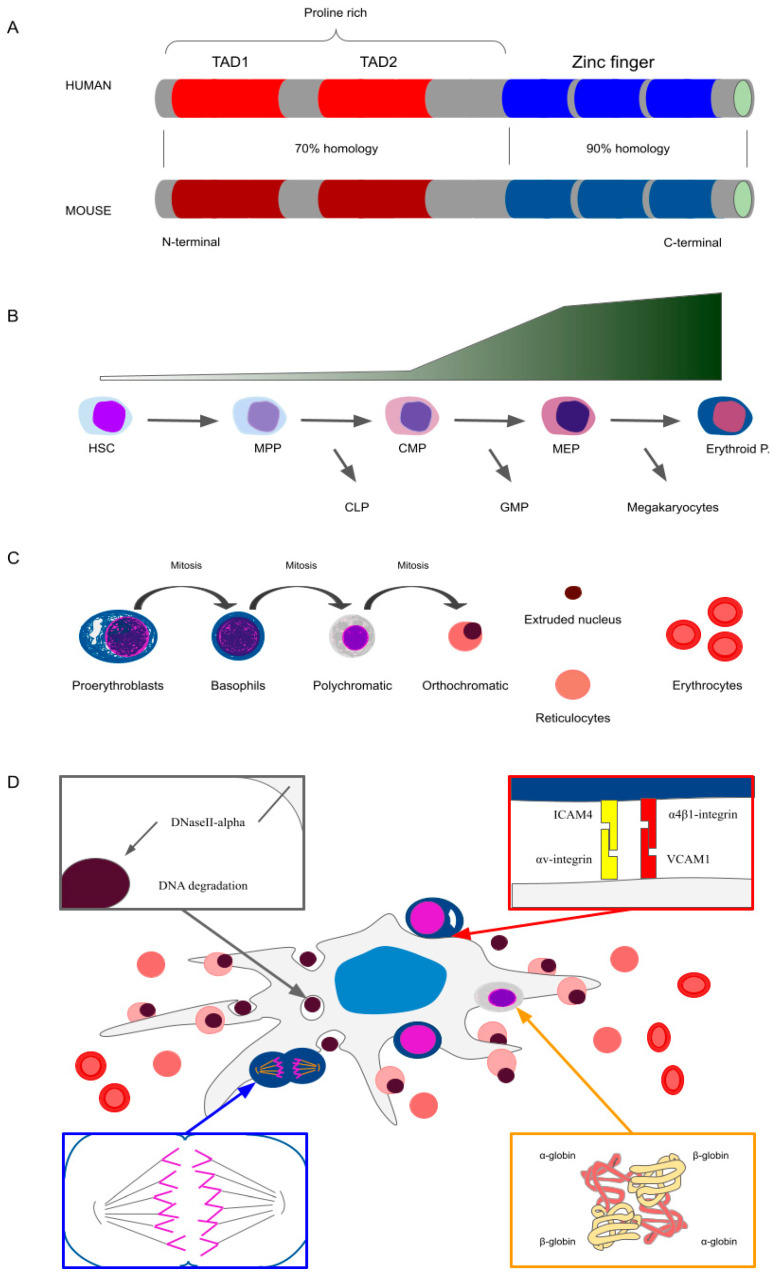
(**A**) Schematic representation of human and mouse KLF1 protein and the relative percentage of domain homology between the two species. TAD: transactivation domain. (**B**) Schematic representation of KLF1 expression levels along erythroid cell commitment from HSCs to fully committed erythroblasts. HSC: hematopoietic stem cells; MPP: multipotent progenitors; CMP: common myeloid progenitors; MEP: megakaryocytic/erythroid progenitors; Erythroid p.: erythroid progenitors; CLP: common lymphoid progenitors; GMP: granulocyte/monocyte progenitors. (**C**) Schematic representation of terminal erythropoiesis. (**D**) Schematic representation of an erythroblastic island. A central macrophage is surrounded by differentiating erythroid cells from proerythroblasts to red blood cells.

**Figure 2 cells-11-03069-f002:**
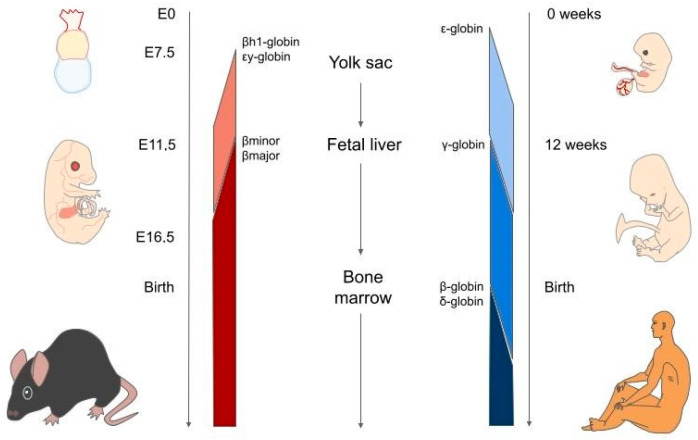
Representation of β-like globin expression in both mice and humans from embryo to adult life. In mice, only one switching event is present at approximately E11.5 (from embryonic to adult globin). In humans, there are two different switching events (from embryonic to foetal globin and from foetal to adult globin) at approximately 12 weeks of intrauterine life and after birth.

**Table 2 cells-11-03069-t002:** Human KLF1 mutations. List of mutations, genotypes and phenotypes described in the present review. These data are sorted by location. Regulatory: the location of the mutation is relative to the transcriptional initiation site. WT: normal allele.

Mutation Type	Nucleotide Change	Predicted Amino Acid Change
Regulatory	-154C > T	
Regulatory	-124T > C	
Nonsense	c.89G > A	Trp30Term
Nonsense	c.172C > T	Gln58Term
Nonsense	c.380T > A	Leu127Term
Missense	c.544T > C	Phe182Leu
Small insertion	c.526_527insCGGCGCC	Gly176AlafsX179
Small insertion	c.519_525dupCGGCGCC	Gly176ArgfsX179
Small deletion	c.569delC	Pro190LeufsX47
Nonsense	c.809C > A	Ser270Term
Nonsense	c.862A > T	Lys288Term
Nonsense	c.874A > T	Lys292Term
Missense	c.892G > C	Ala298Pro
Missense	c.895C > T	His299Tyr
Small insertion	c.954dupG	Arg319GlufsX34
Missense	c.902G > A	Arg301His
Missense	c.973G > A	Glu325Lys
Missense	c.983G > T	Arg328Leu
Missense	c.983G > A	Arg328His
Missense	c.991C > T	Arg331Gly
Missense	c.991C > T	Arg331Trp
Missense	c.994A > C	Lys332Gln
Missense	c.1003G > A	Gly335Arg
Missense	c.1012C > T	Pro338Ser
Missense	c.1012C > A	Pro338Thr
**Monoallelic mutations genotypes-phenotypes**		
**Genotypes**	**Phenotypes**	**References**
Lys292Term/WT	Blood group variant In(Lu)	[27]
Arg319GlufsX34/WT	Blood group variant In(Lu)	[27]
Pro190LeufsX47/WT	Blood group variant In(Lu)	[27]
His299Tyr/WT	Blood group variant In(Lu)	[27]
Arg328Leu/WT	Blood group variant In(Lu)	[27]
Arg328His/WT	Blood group variant In(Lu)	[27]
Arg331Gly/WT	Blood group variant In(Lu)	[27]
Leu127Term/WT	Blood group variant In(Lu)	[27]
-124T>C/WT	Blood group variant In(Lu)	[27]
Lys288Ter/WT	HPFH	[88]
Ser270Term/WT	Increased HbA2 levels	[89]
Glu325Lys/WT	Type IV CDA	[29]
**Compound heterozygous mutations genotypes-phenotypes**		
**Genotypes**	**Phenotypes**	**References**
p.S270X/p.K332Q	HPFH and red cell protoporphyrin	[90]
Trp30Term/Arg319GlufsX34	Hydrops fetalis	[34]
Ala298Pro/Gly176AlafsX179	NSHA	[91]
Pro338Ser/Gly176ArgfsX179	NSHA	[92]
Pro338Thr/Gly176ArgfsX179	NSHA	[93]
Gly335Arg/Gly176ArgfsX179	NSHA	[94]
Arg331Trp/Gly335Arg	NSHA	[95]
Arg301His/Gly176ArgfsX179	NSHA	[94]
-154 C > T/Ala298Pro	NSHA	[95]
Gln58Ter/Ala298Pro	NSHA	[95]
Ala298Pro/Gly176ArgfsX179	NSHA and pyruvate kinase deficiency	[95]
**Homozygous mutation genotype-phenotype**		
**Genotype**	**Phenotype**	**References**
Phe182Leu/Phe182Leu	Increased HbA2 levels	[30]

## Data Availability

Not applicable.

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
