# Peer review of "Krüppel-Like Factor 1: A Pivotal Gene Regulator in Erythropoiesis"

_cells, 2022, doi:10.3390/cells11193069_

Round 1

Reviewer 1 Report

The present review “Krüppel-like factor 1: a pivotal gene regulator in erythropoiesis” by Cristian Caria and co-Authors describes the role of KFL1 in primitive and definitive erythroid cells commitment and its specific roles in the control of the cell cycle, of macrophages function within the erythroblastic island and in the haemoglobin switching. Moreover, it presents an exhaustive description of Klf1 mutations and their pathogenic consequences in human disorders.

The review is timely, complete and well organized. Table1 helps to get oriented in the complex field of Klf1 mutations and variants and it is very useful to readers.

I only have a few suggestions:

I would condense the description of Hematopoiesis (lines68-80) and erythropoiesis (lines 108-127) at the beginning of Section 2 and I would then describe the expression of KLF1 during Hematopoietic lineage commitment and erythroid differentiation. According to this order, in Figure 1 I would move panel 1D just below panel 1B, before describing the more specific cellular KLF1 functions illustrated in panel C.

Lines 269-271: please clarify the nature of the competition for GATA1 binding (at the protein level? in the formation of chromatin-associated complexes? in chromatin occupancy?)

Some Minor inaccuracies and typos are present (see examples here below).

line 325: please add the link to the HGMD database

Line 182: please change “Role of KLF1 in Cell cycle” with “Role of KLF1 in Cell cycle regulation”

Line 214: “its involvement in the regulation of IL12-p40 expression was also demonstrated”

Line 219: “presents”

Line 220: “expresses”

Line 223-224 KLF1- ; KLf1+

Line 385: “É£-promoter”

Line 409: please start a new paragraph from “Finally”

Line 485: “iron levels are normal but iron is not efficiently incorporated”

Author Response

Dear Reviewer,

Thank you for reviewing the manuscript and for all the comments/ suggestions.

Please find enclosed the manuscript amended according to your suggestions.

In addition to the requested changes by the referees, we added a sentence (section 3, lines 365-366) to mention a paper showing that KLF1 directly drives expression of ZBTB7A in erythroid cells, which was missing in the submitted version due to a mere oversight.

In particular, the manuscript has been amended as follows:

  • I would condense the description of Hematopoiesis (lines68-80) and erythropoiesis (lines 108-127) at the beginning of Section 2 and I would then describe the expression of KLF1 during Hematopoietic lineage commitment and erythroid differentiation. According to this order, in Figure 1 I would move panel 1D just below panel 1B, before describing the more specific cellular KLF1 functions illustrated in panel C.

According to your suggestion, we moved lines 68-80 to the beginning of section 2, condensing haematopoiesis and erythropoiesis description. Figure 1 has been changed by moving panel 1D below panel 1B. Numbering, reference in the text and legenda have been modified in agreement.

  • Lines 269-271: please clarify the nature of the competition for GATA1 binding (at the protein level? in the formation of chromatin-associated complexes? in chromatin occupancy?)

The protein-protein nature of competition for GATA1 binding was stated.

  • line 325: please add the link to the HGMD database

The link was added

  • Line 182: please change “Role of KLF1 in Cell cycle” with “Role of KLF1 in Cell cycle regulation”

The phrase was changed

  • Line 214: “its involvement in the regulation of IL12-p40 expression was also demonstrated”

The phrase was corrected

  • Line 219: “presents”

The phrase was corrected 

  • Line 220: “expresses”

The phrase was corrected

  • Line 223-224 KLF1-; KLf1+

The phrase was corrected

  • Line 385: “É£-promoter

The phrase was corrected

  • Line 409: please start a new paragraph from “Finally”

A new paragraph was created

  • Line 485: “iron levels are normal but iron is not efficiently incorporated”

The phrase was corrected.

The paper has been carefully revised for any minor inaccuracies and typos.

Sincerely

Maria Serafina Ristaldi

Reviewer 2 Report

This is a comprehensive review of genetical and biological functions of KLF1 in erythropoiesis. Thus, this review is very interesting and worthy for future progress for medical science for red blood cell diseases. I recommend some minor revise to improve this review more valuable.

Comments: 

1. For showing pleiotropic function of KLF1, it is better to show target genes in a table. 

2. Lines 99-106: It is a too large legend. Main part of sentences can be moved to discussion.

3. Papers on KLF1 were published in recent 5 years (after 2018). Please include newer references if possible.

4. Recent development in erythropoiesis contained epigenetic control of erythroid gene expression, such as miRNAs, CRISPR technology. If authors include these topics, tthe review may become more attractive.

5. I am interested in overexpression of KLF1 in central macrophages. Please give some comments on the mechanism of the expression of erythroid-specific gene: KLF1.

6. I think there is confusion of KLF1 function between beta-thalassemia and Hb switching. Beta-thalassemia is caused by genetic abnormalities of beta-globin gene, which is different from the mechanism of Hb switching. 

7. Related comments on some confusion between erythropoiesis and Hb switching.

Author Response

Dear Reviewer,

Thank you for reviewing the manuscript and for all the comments/ suggestions.

Please find enclosed the manuscript amended according to your suggestions.

In addition to the requested changes by the referees, we added a sentence (section 3, lines 365-366) to mention a paper showing that KLF1 directly drives expression of ZBTB7A in erythroid cells, which was missing in the submitted version due to a mere oversight.

In particular, the manuscript has been amended as follows:

  1. For showing pleiotropic function of KLF1, it is better to show target genes in a table. 

A new table (Table 1) was created containing the genes influenced by KLF1 described in the manuscript.

  1. Lines 99-106: It is a too large legend. Main part of sentences can be moved to discussion.

We have eliminated parts of the legend which is now more concise.

  1. Papers on KLF1 were published in recent 5 years (after 2018). Please include newer references if possible.

We apologize for the bibliographic gaps in manuscript. We have updated the references.

  1. Recent development in erythropoiesis contained epigenetic control of erythroid gene expression, such as miRNAs, CRISPR technology. If authors include these topics, the review may become more attractive.

We thank you for the suggestion. We have added a new section on epigenetic regulation of KLF1 by ncRNAs.

  1. I am interested in overexpression of KLF1 in central macrophages. Please give some comments on the mechanism of the expression of erythroid-specific gene: KLF1.

We added a paragraph describing the effects of KLF1 overexpression in macrophages in section 2.2.2. Moreover, we inserted a comment on the erythroid specific expression in section 2.

  1. I think there is confusion of KLF1 function between beta-thalassemia and Hb switching. Beta-thalassemia is caused by genetic abnormalities of beta-globin gene, which is different from the mechanism of Hb switching. 

We apologize for any confusion in the manuscript. We have revised some paragraphs with the aim of avoiding misunderstanding between beta-thalassemia and Hb switching. All changes are evidenced in the new uploaded version of the manuscript.

  1. Related comments on some confusion between erythropoiesis and Hb switching.

As in the previous point, we hope we have solved this problem.

Sincerely

Maria Serafina Ristaldi

Reviewer 3 Report

This is a comprehensive review with some particularly valuable sections that have not been covered by previous reviews of KLF1. There a few English grammar errors and some additional references and changes that would improve the review. I have listed them below by line

Line 2 - title. I think it should be 'pivotal' rather than "pivot'

Line 38 - change 'function as" to 'recruit"

Line 43 - it is still controversial as to whether KLF1 has repressor activity and this was not claimed in ref 2. I think an easy way to avoid the controversy would be to change the wording from 'KLF1 also performs repressor activity, albeit to a lessor extent' to "KLF1 may act as a repressor in some contexts'. I would remove the sentence on lines 44-45 starting with 'This dual function...

Line 60 'KLF1 null embryo fetal livers are normal in size' Although they are normal in size at E13.5 they are actually about two thirds of normal size at E14.5 which reflects defective erythropoiesis. I suggest using the phrase 'near-normal'. (see Ref 12 fig 1 for a good example of the fetal liver size).  

Lines 66 'metabolism and structure proteins, through the formation....." This is poorly written. I think a change to 'metabolism and structure, and the formation...."

Lines 68-75 and Figure 1B. I have some concerns about this section that is primarily based on citation 12. Although this is an interesting paper, the expression of KLF1 in stem and progenitor cells is based on FACs and even slight contamination with erythroid cells will alter the apparent expression. Other look up references for this (e.g. Haemosphere) and single cell RNA-seq data sets suggest KLF1 is extremely lowly expressed in HSCs and all progenitors prior to the MEP stage when it rapidly increases. I suggest keeping reference 12 but simplifying the section from lines 68-74 to simply say 'KLF1 is expressed at low levels in HSCs and progenitor cells and first detected at reasonable levels in MEPs" I think Fig 2B needs to change also to reflect this rather than a bi-phasic expression trajectory. 

Line 79 - change 'highly' to 'markedly' 

Page 7, section 2.2.1. The references at the end of line 4 should include the first description of KLF1 regulation of a cell cycle gene, p18. (Tallack et al. J Mol Biol. 2007 Jun 1;369(2):313-21). 

Page 8, first paragraph. The same reference above should be added to the references at the end of line 8, after 'p18, p21 and p27'. 

Page 8, section 2.2.2 - I enjoyed this section very much

However, line 15, change 'indicated' to 'identified' 

Page 8, last four lines. The English here is a little awkward. I would change 'If a lack....' To "Lack of KLF1 results in downregulation of ICAM4.....in erythroid cells.....

Page 9 section 2.2.3, line 10

Change 'Fli-1 occupancy on platelets restricted genes, thus promoting megakaryocytic differentiation' to "Fli-1 occupancy at promoters and enhancers of genes that promote megakaryocytic differentiation"

Page 9 last line continuing to page 10 first paragraph

As discussed above the is based on reference 12 only. There is other evidence that Klf1 null HSCs, myeloid progenitors and white blood cells (perhaps with the exclusion of CMBIs) are all normal in transplanted mice (see Tallack et al. Haematologica. 2010 Jan;95(1):144-7)

I think it is easiest just to delete this whole section from the forth bottom line on page 8 to the end of the first paragraph of page 10 rather than provide conflicting evidence. It is very doubtful there is any significant role for KLF1 in the HSC. 

Page 10 section 3, paragraph 2, line 2. Change KLF1-/- to Klf1-/- to be consistent with nomenclature guidelines. This should be carefully checked throughout....i.e. when referring to mouse or human and gene or protein. 

Line 6. An addition reference to the equal first observation of the role of KLF1 in human globin gene switching in bYAC mice from the Orkin laboratory. This paper clearly hypothesised on the competition model that was novel at the time (Perkins et al. Proc Natl Acad Sci U S A. 1996 Oct 29;93(22):12267-71). It might be worth mentioning here the recent work from the Crossley laboratory that shows that damaging the KLF1-binding site in the beta-globin promoter is sufficient to disable the normal competition for the LCR and reactivate fetal hemoglobin (i.e. independent of any action of BCL11A). (Topfer SK et al. Blood. 2022 Apr 7;139(14):2107-2118). 

Page 14, lines 10-11. "This finding suggests that genetic modifers...." I dont think this is a required conclusion from the previous section. I think the elevation or not of HbF in KLF1 mutant cases can easily be explained by the degree to which mutations are full loss of function mutations or hypomorphic variants (as in Lys332Gln). 

Page 14 end of section 1. It might be nice to include mention of a forgotten gene, AHSP, here. AHSP is a chaparone for unfolded and excess a-globin chains (Yu et al. J Clin Invest. 2007 Jul;117(7):1856-65). It is a direct target gene of KLF1 and loss of expression in KLF1 mutant cells exacerbates the damage conferred by the excess of a-globin chains (Keys et al. Br J Haematol. 2007 Jan;136(1):150-7). 

Page 15 end of paragraph 1. There are some new papers that look at the consequences of the nan mutation on the erythroid transcriptome in detail. I suggest the authors include (with ref 95) reference to Nebor et al. Sci Rep. 2018 Aug 24;8(1):12793 and/or Planutis A. et al. Development. 2017 Feb 1;144(3):430-440

Also at the end of the paragraph there needs to be an explanation as to why the dominant mutations at the central DNA-contact residue in ZF2 cause dominant disease via a neo-function. This is why only mutations at this one site in KLF1 are autosomal dominant. It is important for genetic counselling. The relevant papers on this are Nebor et al. (see above), Gillinder KR, et al. Nucleic Acids Res. 2017 Feb 17;45(3):1130-1143; Ilsley MD et al. BMC Genomics. 2019 May 24;20(1):417; and Kulczynska-Figurny K. et al. Mutat Res Rev Mutat Res. 2020 Oct-Dec;786:108336

Page 16, paragraph 2, line 3. Can the authors add ref 3 to refs 29, 30 and 99.

In summary, this is an excellent broad review of KLF1's role in erythropoiesis. It draws on and integrates work from mouse and human studies. It will be of broad interest to pediatric haematologist that find KLF1 variants in their patients and to those interested in how transcription factors regulate development and differentiation of the erythroid lineage. 

Author Response

Dear Reviewer,

Thank you for reviewing the manuscript and for all the comments/ suggestions.

Please find enclosed the manuscript amended according to your suggestions.

In addition to the requested changes by the referees, we added a sentence (section 3, lines 365-366) to mention a paper showing that KLF1 directly drives expression of ZBTB7A in erythroid cells, which was missing in the submitted version due to a mere oversight.

In particular, the manuscript has been amended as follows:

Line 2 - title. I think it should be 'pivotal' rather than "pivot'

We agree and have proceeded to change the title

Line 38 - change 'function as" to 'recruit"

We have changed 'function as" to 'recruit" as suggested.

Line 43 - it is still controversial as to whether KLF1 has repressor activity and this was not claimed in ref 2. I think an easy way to avoid the controversy would be to change the wording from 'KLF1 also performs repressor activity, albeit to a lessor extent' to "KLF1 may act as a repressor in some contexts'. I would remove the sentence on lines 44-45 starting with 'This dual function...

We agree and have proceeded to change the sentence as suggested.

Line 60 'KLF1 null embryo fetal livers are normal in size' Although they are normal in size at E13.5 they are actually about two thirds of normal size at E14.5 which reflects defective erythropoiesis. I suggest using the phrase 'near-normal'. (see Ref 12 fig 1 for a good example of the fetal liver size).  

We agree and provide to change the sentence as suggested.

Lines 66 'metabolism and structure proteins, through the formation....." This is poorly written. I think a change to 'metabolism and structure, and the formation...."

We apologize for the poorly written sentence. We provided to change it.

Lines 68-75 and Figure 1B. I have some concerns about this section that is primarily based on citation 12. Although this is an interesting paper, the expression of KLF1 in stem and progenitor cells is based on FACs and even slight contamination with erythroid cells will alter the apparent expression. Other look up references for this (e.g. Haemosphere) and single cell RNA-seq data sets suggest KLF1 is extremely lowly expressed in HSCs and all progenitors prior to the MEP stage when it rapidly increases. I suggest keeping reference 12 but simplifying the section from lines 68-74 to simply say 'KLF1 is expressed at low levels in HSCs and progenitor cells and first detected at reasonable levels in MEPs" I think Fig 1B needs to change also to reflect this rather than a bi-phasic expression trajectory. 

We acknowledged your suggestion and proceeded to change both the sentence and the figure.

Line 79 - change 'highly' to 'markedly' 

The phrase was changed

Page 7, section 2.2.1. The references at the end of line 4 should include the first description of KLF1 regulation of a cell cycle gene, p18. (Tallack et al. J Mol Biol. 2007 Jun 1;369(2):313-21). 

We added the reference

Page 8, first paragraph. The same reference above should be added to the references at the end of line 8, after 'p18, p21 and p27'. 

We added the reference

Page 8, section 2.2.2 - I enjoyed this section very much

We are glad you liked the section. Thank you very much.

However, line 15, change 'indicated' to 'identified' 

The phrase was changed

Page 8, last four lines. The English here is a little awkward. I would change 'If a lack....' To "Lack of KLF1 results in downregulation of ICAM4.....in erythroid cells.....

We apologize for the unintelligible sentence. We provided to change the phrase as suggested.

Page 9 section 2.2.3, line 10

Change 'Fli-1 occupancy on platelets restricted genes, thus promoting megakaryocytic differentiation' to "Fli-1 occupancy at promoters and enhancers of genes that promote megakaryocytic differentiation"

The phrase was changed as suggested.

Page 9 last line continuing to page 10 first paragraph

As discussed above the is based on reference 12 only. There is other evidence that Klf1 null HSCs, myeloid progenitors and white blood cells (perhaps with the exclusion of CMBIs) are all normal in transplanted mice (see Tallack et al. Haematologica. 2010 Jan;95(1):144-7)

I think it is easiest just to delete this whole section from the forth bottom line on page 8 to the end of the first paragraph of page 10 rather than provide conflicting evidence. It is very doubtful there is any significant role for KLF1 in the HSC. 

We have followed your suggestion and deleted the paragraph.

Page 10 section 3, paragraph 2, line 2. Change KLF1-/- to Klf1-/- to be consistent with nomenclature guidelines. This should be carefully checked throughout....i.e. when referring to mouse or human and gene or protein. 

 We apologize for this. We have carefully checked the manuscript throughout and followed the nomenclature guidelines in this revised version of the manuscript.

Line 6. An addition reference to the equal first observation of the role of KLF1 in human globin gene switching in bYAC mice from the Orkin laboratory. This paper clearly hypothesised on the competition model that was novel at the time (Perkins et al. Proc Natl Acad Sci U S A. 1996 Oct 29;93(22):12267-71). It might be worth mentioning here the recent work from the Crossley laboratory that shows that damaging the KLF1-binding site in the beta-globin promoter is sufficient to disable the normal competition for the LCR and reactivate fetal hemoglobin (i.e. independent of any action of BCL11A). (Topfer SK et al. Blood. 2022 Apr 7;139(14):2107-2118). 

We thank you for the comprehensive explanation and bibliographic suggestion. We provided to mention Crossley laboratory’s work in the manuscript.

Page 14, lines 10-11. "This finding suggests that genetic modifers...." I dont think this is a required conclusion from the previous section. I think the elevation or not of HbF in KLF1 mutant cases can easily be explained by the degree to which mutations are full loss of function mutations or hypomorphic variants (as in Lys332Gln). 

We have removed the sentence you pointed out as incorrect.

Page 14 end of section 1. It might be nice to include mention of a forgotten gene, AHSP, here. AHSP is a chaparone for unfolded and excess a-globin chains (Yu et al. J Clin Invest. 2007 Jul;117(7):1856-65). It is a direct target gene of KLF1 and loss of expression in KLF1 mutant cells exacerbates the damage conferred by the excess of a-globin chains (Keys et al. Br J Haematol. 2007 Jan;136(1):150-7). 

We have now mentioned AHSP gene in the paper (section 5.4, lines 537-542) and added suggested references.

Page 15 end of paragraph 1. There are some new papers that look at the consequences of the nan mutation on the erythroid transcriptome in detail. I suggest the authors include (with ref 95) reference to Nebor et al. Sci Rep. 2018 Aug 24;8(1):12793 and/or Planutis A. et al. Development. 2017 Feb 1;144(3):430-440

We added the suggested references.

Also at the end of the paragraph there needs to be an explanation as to why the dominant mutations at the central DNA-contact residue in ZF2 cause dominant disease via a neo-function. This is why only mutations at this one site in KLF1 are autosomal dominant. It is important for genetic counselling. The relevant papers on this are Nebor et al. (see above), Gillinder KR, et al. Nucleic Acids Res. 2017 Feb 17;45(3):1130-1143; Ilsley MD et al. BMC Genomics. 2019 May 24;20(1):417; and Kulczynska-Figurny K. et al. Mutat Res Rev Mutat Res. 2020 Oct-Dec;786:108336

 We thank you for the excellent advice. We have improved the paragraph following your suggestions.

Page 16, paragraph 2, line 3. Can the authors add ref 3 to refs 29, 30 and 99.

We added the suggested references.

Sincerely

Maria Serafina Ristaldi